# COMPETITIVE PHYSICS INFORMED NETWORKS

**Qi Zeng, Yash Kothari, Spencer H. Bryngelson & Florian Schäfer**
School of Computational Science and Engineering
Georgia Institute of Technology
Atlanta, GA 30332, USA
`{qzeng37@,ykothari3@,shb@,florian.schaefer@cc.}gatech.edu`

## ABSTRACT

Neural networks can be trained to solve partial differential equations (PDEs) by using the PDE residual as the loss function. This strategy is called "physics-informed neural networks" (PINNs), but it currently cannot produce high-accuracy solutions, typically attaining about $0.1\%$ relative error. We present an adversarial approach that overcomes this limitation, which we call competitive PINNs (CPINNs). CPINNs train a discriminator that is rewarded for predicting mistakes the PINN makes. The discriminator and PINN participate in a zero-sum game with the exact PDE solution as an optimal strategy. This approach avoids squaring the large condition numbers of PDE discretizations, which is the likely reason for failures of previous attempts to decrease PINN errors even on benign problems. Numerical experiments on a Poisson problem show that CPINNs achieve errors four orders of magnitude smaller than the best-performing PINN. We observe relative errors on the order of single-precision accuracy, consistently decreasing with each epoch. To the authors' knowledge, this is the first time this level of accuracy and convergence behavior has been achieved. Additional experiments on the nonlinear Schrödinger, Burgers', and Allen–Cahn equation show that the benefits of CPINNs are not limited to linear problems.

## 1 INTRODUCTION

**PDE-constrained deep learning.** Partial differential equations (PDEs) model physical phenomena like fluid dynamics, heat transfer, electromagnetism, and more. The rising interest in scientific machine learning motivates the study of PDE-constrained neural network training (Lavin et al., 2021). Such methods can exploit physical structure for learning or serve as PDE solvers in their own right.

**Physics informed networks.** Lagaris et al. (1998) represent PDE solutions as neural networks by including the square of the PDE residual in the loss function, resulting in a neural network-based PDE solver. Raissi et al. (2019) recently refined this approach further and called it "physics informed neural networks (PINNs)," initiating a flurry of follow-up work. PINNs are *far* less efficient than classical methods for solving most PDEs but are promising tools for high-dimensional or parametric PDEs (Xue et al., 2020) and data assimilation problems. The training of PINNs also serves as a model problem for the general challenge of imposing physical constraints on neural networks, an area of fervent and increasing interest (Wang et al., 2021b; Li et al., 2021; Donti et al., 2021).

**Training pathologies in PINNs.** PINNs can, in principle, be applied to all PDEs, but their numerous failure modes are well-documented (Wang et al., 2021a; Liu et al., 2021; Krishnapriyan et al., 2021). For example, they are often unable to achieve high-accuracy solutions. The first works on PINNs reported relative $L_2$ errors of about $10^{-3}$ (Raissi et al., 2019). The authors are unaware of PINNs achieving errors below $10^{-5}$, even in carefully crafted, favorable settings. Higher accuracy is required in many applications.

**Existing remedies.** A vast and growing body of work aims to improve the training of PINNs, often using problem-specific insights. For example, curriculum learning can exploit causality in time-dependent PDEs (Krishnapriyan et al., 2021; Wang et al., 2022a; Wight & Zhao, 2020). Krishnapriyan et al. (2021) also design curricula by embedding the PDE in a parametric family of problems of varying difficulty. Other works propose adaptive methods for selecting the PINN collocation points (Lu et al.,

2021; Nabian et al., 2021; Daw et al., 2022). Adaptive algorithms for weighing components of the PINN loss function have also been proposed (McClenny & Braga-Neto, 2020; Wang et al., 2022b; van der Meer et al., 2022). Despite these improvements, the squared residual penalty method used by such PINNs imposes a fundamental limitation associated with conditioning, which is discussed next.

**The key problem: Squared residuals.** Virtually all PINN-variants use the squared PDE residual as loss functions. For a linear PDE of order $s$, this is no different than solving an equation of order $2s$, akin to using normal equations in linear algebra. The condition number $\kappa$ of the resulting problem is thus the *square* of the condition number of the original one. Solving discretized PDEs is an ill-conditioned problem, inhibiting the convergence of iterative solvers and *explaining the low accuracy of most previous PINNs*. It is tempting to address this problem using penalties derived from $p$-norms with $p \neq 2$. However, choosing $p > 2$ leads to worse condition numbers, whereas $p < 2$ sacrifices the smoothness of the objective. The convergence rates of gradient descent on (non-)smooth convex problems suggest that this trade is unfavorable (Bubeck et al., 2015).

**Weak formulations.** Integration by parts allows the derivation of a weak form of a PDE, which for some PDEs can be turned into a minimization formulation that does not square the condition number. This procedure has been successfully applied by E & Yu (2017) to solve PDEs with neural networks (*Deep Ritz*). However, the derivation of such minimization principles is problem-dependent, limiting the generality of the formulation. Deep Ritz also employs penalty methods to enforce boundary values, though these preclude the minimization problem's solution from being the PDE's exact solution. Liao & Ming (2019) proposed a partial solution to this problem. The work most closely related to ours is by Zang et al. (2020), who proposes a game formulation based on the weak form.

**Competitive PINNs.** We propose *Competitive* Physics Informed Neural Networks (CPINNs) to address the above problems. CPINNs are trained using a minimax game between the PINN and a discriminator network. The discriminator learns to predict mistakes of the PINN and is rewarded for correct predictions, whereas the PINN is penalized. We train both players simultaneously on the resulting zero-sum game to reach a Nash equilibrium that matches the exact solution of the PDE.

**Summary of contributions.** A novel variant of PINNs, called CPINNs, is introduced, replacing the penalty employed by PINNs with a primal-dual approach. This simultaneously optimizes the PINN and a discriminator network that learns to identify PDE violations and boundary constraint violations. We optimize PINNs with competitive gradient (CGD) (Schäfer & Anandkumar, 2019) and compare their performance to regular PINNs trained with Adam. On a two-dimensional Poisson problem, CPINNs achieve a relative accuracy of almost $10^{-8}$, improving over PINNs by four orders of magnitude. To the best of our knowledge, this is the first time a PINN-like network was trained to this level of accuracy. We compare PINNs with CPINNs on a nonlinear Schrödinger equation, a viscous Burgers' equation, and an Allen-Cahn equation. In all but the last case, CPINNs improve over PINNs trained with a comparable computational budget.

## 2 COMPETITIVE PINN FORMULATION

We formulate CPINNs for a PDE of the general form

$$\mathcal{A}[u] = f, \quad \text{in } \Omega \tag{1}$$
$$u = g, \quad \text{on } \partial\Omega, \tag{2}$$

where $\mathcal{A}[\cdot]$ is a (possibly nonlinear) differential operator and $\Omega$ is a domain in $\mathbb{R}^d$ with boundary $\partial\Omega$. To simplify notation, we assume that $f$, $g$, and $u$ are real-valued functions on $\Omega$, $\partial\Omega$, and $\Omega \cup \partial\Omega$, respectively. One can extend both PINNs and CPINNs to vector-valued such functions if needed.

### 2.1 PHYSICS INFORMED NEURAL NETWORKS (PINNS)

PINNs approximate the PDE solution $u$ by a neural network $\mathcal{P}$ mapping $d$-variate inputs to real numbers. The weights are chosen such as to satisfy equation 1 and equation 2 on the points $\boldsymbol{x} \subset \Omega$ and $\overline{\boldsymbol{x}} \subset \partial\Omega$. The loss function used to train $\mathcal{P}$ has the form

$$\mathcal{L}^{\text{PINN}}(\mathcal{P}, \boldsymbol{x}, \overline{\boldsymbol{x}}) = \mathcal{L}^{\text{PINN}}_{\Omega}(\mathcal{P}, \boldsymbol{x}_{\Omega}) + \mathcal{L}^{\text{PINN}}_{\partial\Omega}(\mathcal{P}, \overline{\boldsymbol{x}}), \tag{3}$$

where $\mathcal{L}_{\partial\Omega}^{\text{PINN}}$ measures the violation of the boundary conditions equation 2 and $\mathcal{L}_{\Omega}^{\text{PINN}}$ measures the violation of the PDE of equation 1. They are defined as

$$\mathcal{L}_{\Omega}^{\text{PINN}}(\mathcal{P}, \boldsymbol{x}) = \frac{1}{N_{\Omega}} \sum_{i=1}^{N_{\Omega}} \left( \mathcal{A}[\mathcal{P}](x_i)) - f(x_i) \right)^2 \tag{4}$$

$$\mathcal{L}_{\partial\Omega}^{\text{PINN}}(\mathcal{P}, \overline{\boldsymbol{x}}) = \frac{1}{N_{\partial\Omega}} \sum_{i=1}^{N_{\partial\Omega}} \left( \mathcal{P}\left(\overline{x}_i\right) - g\left(\overline{x}_i\right) \right)^2. \tag{5}$$

Here, $N_{\Omega}$ and $N_{\partial\Omega}$ are the number of points in the sets $\boldsymbol{x}$ (interior) and $\overline{\boldsymbol{x}}$ (boundary), and $x_i$ and $\overline{x}_i$ are the $i$-th such points in $\boldsymbol{x}$ and $\overline{\boldsymbol{x}}$.

PINNs approximate the exact solution $u$ of the PDE by minimizing the loss in equation 3. However, optimizing this loss using established methods such as gradient descent, ADAM, or LBFGS often leads to unacceptably large errors or an inability to train at all (Wang et al., 2021a; 2022b). This pathology has been attributed to the bad conditioning of the training problem (Krishnapriyan et al., 2021). Next, we introduce CPINNs, a game-based formulation designed to mitigate these problems.

## 2.2 Competitive Physics Informed Neural Networks (CPINNs)

CPINNs introduce one or more discriminator networks $\mathcal{D}$ with input $x \in \mathbb{R}^d$ and outputs $\mathcal{D}_{\Omega}(x)$ and $\mathcal{D}_{\partial\Omega}(x)$. $\mathcal{P}$ and $\mathcal{D}$ compete in a zero-sum game where $\mathcal{P}$ learns to solve the PDE, and $\mathcal{D}$ learns to predict the mistakes of $\mathcal{P}$. This game is defined as a minimax problem

$$\max_{\mathcal{D}} \min_{\mathcal{P}} \mathcal{L}_{\Omega}^{\text{CPINN}}(\mathcal{P}, \mathcal{D}, \boldsymbol{x}) + \mathcal{L}_{\partial\Omega}^{\text{CPINN}}(\mathcal{P}, \mathcal{D}, \overline{\boldsymbol{x}}), \tag{6}$$

where

$$\mathcal{L}_{\Omega}^{\text{CPINN}}(\mathcal{D}, \mathcal{P}, \boldsymbol{x}) = \frac{1}{N_{\Omega}} \sum_{i=1}^{N_{\Omega}} \mathcal{D}_{\Omega}(x_i) \left( \mathcal{A}[\mathcal{P}](x_i) - f(x_i) \right), \tag{7}$$

$$\mathcal{L}_{\partial\Omega}^{\text{CPINN}}(\mathcal{D}, \mathcal{P}, \overline{\boldsymbol{x}}) = \frac{1}{N_{\partial\Omega}} \sum_{i=1}^{N_{\partial\Omega}} \mathcal{D}_{\partial\Omega}(\overline{x}_i) \left( \mathcal{P}\left(\overline{x}_i\right) - g\left(\overline{x}_i\right) \right). \tag{8}$$

Here, $\mathcal{D}_{\Omega}(x_i)$ and $\mathcal{D}_{\partial\Omega}(\overline{x})$ can be interpreted as *bets* by the discriminator that the PINN will over- or under-shoot equation 1 and equation 2. Winning the bet results in a reward for $\mathcal{D}$ and a penalty for $\mathcal{P}$; a lost bet has the opposite effect. We emphasize that choosing bets on the residual is fundamentally different from choosing weights for the different components of a loss function, as done by McClenny & Braga-Neto (2020); Wang et al. (2022b); van der Meer et al. (2022)

The Nash equilibrium of this game is $\mathcal{P} \equiv u$ and $\mathcal{D} \equiv 0$. Thus, iterative algorithms for computing Nash equilibria in such zero-sum games can be used to solve the PDE approximately. This work focuses on the CGD algorithm of Schäfer & Anandkumar (2019). Still, CPINNs can be trained with other proposed methods for smooth game optimization (Korpelevich, 1977; Mescheder et al., 2017; Balduzzi et al., 2018; Gemp & Mahadevan, 2018; Daskalakis et al., 2018; Letcher et al., 2019). In our experiments, $\mathcal{P}$ and $\mathcal{D}$ are fully connected networks with hyperbolic tangent and ReLU activation functions, respectively. Each network's number of layers and neurons depends on the PDE problem.

## 2.3 Avoiding squares of differential operators

Multiagent methods for solving PDEs may seem unorthodox, yet they are motivated by observations in classical numerical analysis. Consider the particular case of a linear PDE and networks $\mathcal{P}, \mathcal{D}$ with outputs that depend linearly on their weight-vectors $\boldsymbol{\pi}$ and $\boldsymbol{\delta}$ resulting in the parametric form

$$\mathcal{P}(x) = \sum_{i=1}^{\dim(\boldsymbol{\pi})} \pi_i \psi_i(x), \quad \mathcal{D}(x) = \sum_{i=1}^{\dim(\boldsymbol{\delta})} \delta_i \phi_i(x), \tag{9}$$

for basis function sets $\{\psi_i\}_{1 \le i \le \dim(\boldsymbol{\pi})}$ and $\{\phi_i\}_{1 \le i \le \dim(\boldsymbol{\delta})}$. We focus our attention on the PDE constraint in equation 1, which is evaluated at a set $\boldsymbol{x}$ of $N_{\Omega}$ points. Defining $\boldsymbol{A} \in \mathbb{R}^{N_{\Omega} \times \dim(\boldsymbol{\pi})}$ and $\boldsymbol{f} \in \mathbb{R}^{N_{\Omega}}$, this leads to

$$A_{ij} := \mathcal{A}[\psi_j](x_i), \quad f_i := f(x_i) \tag{10}$$

and the discretized PDE

$$A\pi = f. \tag{11}$$

PINNs solve equation 11 via a least squares problem

$$\min_{\pi} \|A\pi - f\|^2, \tag{12}$$

trading the equality constraint for a minimization problem with solution $\pi = (A^\top A)^{-1} A^\top f$.

Since the matrix $(A^\top A)$ is symmetric positive-definite, one can solve it with specialized algorithms such as the conjugate gradient method (CG) (Shewchuk, 1994). This approach is beneficial for well-conditioned nonsymmetric matrices but inappropriate for ill-conditioned $A$ (Axelsson, 1977). This is because $\kappa(A^\top A) = \kappa(A)^2$, resulting in slow convergence of iterative solvers. Differential operators are unbounded. Thus, their discretization leads to ill-conditioned linear systems. Krishnapriyan et al. (2021) argue that the ill-conditioning of equation 12 causes the optimization difficulties they observe.

CPINNs turn the discretized PDE in equation 11 into the saddlepoint problem

$$\min_{\pi} \max_{\delta} \delta^\top (A\pi - f). \tag{13}$$

The solution to this problem is the same as that of the system of equations

$$\begin{bmatrix} 0 & A^\top \\ A & 0 \end{bmatrix} \begin{bmatrix} \pi \\ \delta \end{bmatrix} = \begin{bmatrix} 0 \\ f \end{bmatrix}, \quad \text{with} \quad \kappa\left(\begin{bmatrix} 0 & A^\top \\ A & 0 \end{bmatrix}\right) = \kappa(A). \tag{14}$$

By turning equation 11 into the saddle point problem of equation 13 instead of the minimization form of equation 12, CPINNs avoid squaring the condition number. Applying a Krylov subspace method directly to equation 14 will not improve the convergence compared to working with $A^\top A$. However, gradients of equation 13 provide access to matrix-vector products with $A$ and $A^\top$, which can not be obtained from gradients of the least-squares problem. For many matrices, including symmetric positive-definite ones, this allows the significantly faster solution of equation 11 (Greenbaum, 1997). We now demonstrate empirically the benefits of equation 13 in the nonlinear setting.

## 3 RESULTS

### 3.1 OVERVIEW

This section compares PINNs and CPINNs on a series of model problems. The code used to produce the experiments described below can be found under `github.com/comp-physics/CPINN`. We study a two-dimensional Poisson problem (section 3.2), a nonlinear Schrödinger equation (section 3.3), a viscous Burgers' equation (section 3.4), and the Allen-Cahn equation (section 3.5). Throughout the previous sections, we train PINNs using Adam and CPINN using adaptive competitive gradient descent (ACGD) (Schäfer et al., 2020b). The latter combines CGD of Schäfer & Anandkumar (2019) with an Adam-like heuristic for choosing step sizes.

ACGD uses GMRES (Saad & Schultz, 1986) and the Hessian vector products obtained by automatic differentiation to solve the linear system defining the CGD update. Thus, iterations of (A)CGD are considerably more expensive than those of Adam. To account for this difference fairly, we also provide the number of forward passes through the neural network required by the two methods. An Adam iteration amounts to a single forward pass and an ACGD iteration to two forward passes, plus two times the number of GMRES iterations. We also tried other optimizers for both PINNs and CPINNs but found them inferior to Adam and ACGD. A comparison is presented in section 3.6.

### 3.2 POISSON EQUATION

We begin by considering a two-dimensional Poisson equation:

$$\Delta u(x,y) = -2\sin(x)\cos(y), \quad x,y \in [-2,2] \tag{15}$$

with Dirichlet boundary conditions

$$\begin{aligned} u(x,-2) &= \sin(x)\cos(-2), & u(-2,y) &= \sin(-2)\cos(y), \\ u(x,\phantom{-}2) &= \sin(x)\cos(\phantom{-}2), & u(\phantom{-}2,y) &= \sin(\phantom{-}2)\cos(y). \end{aligned}$$

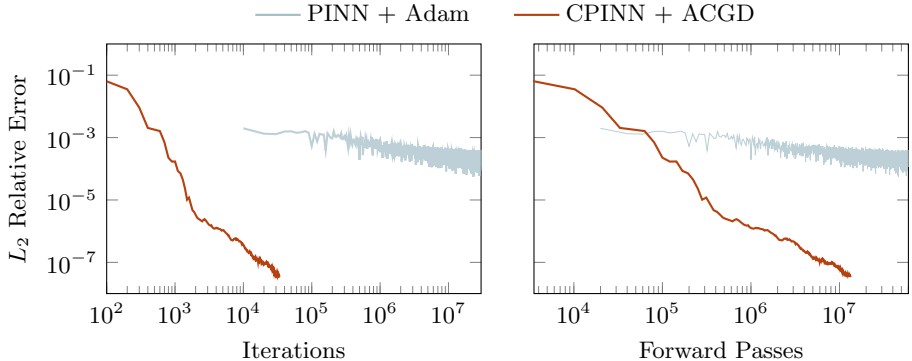

Figure 1: Comparison of CPINN and PINN on the Poisson problem of equation 15 in terms of relative error. CPINN has a faster convergence rate and reduces the $L_2$ error to $1.7 \times 10^{-8}$, whereas the PINN case has an $L_2$ error of $1.2 \times 10^{-4}$ even with a larger computational budget.

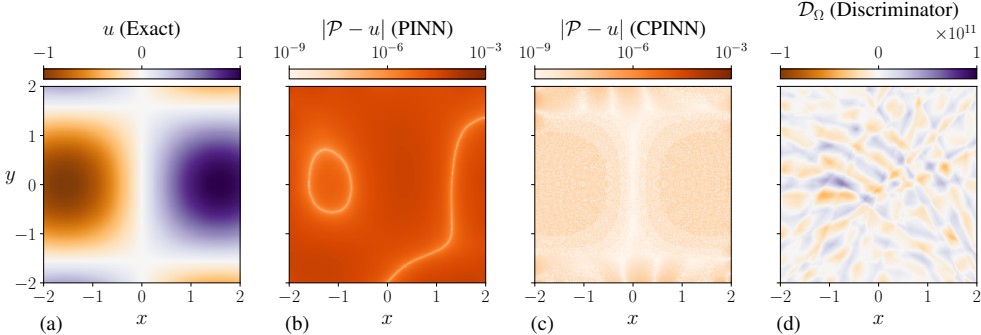

Figure 2: (a) Exact solution $u$ to equation 1, absolute errors of (b) PINN + Adam after $3 \times 10^7$ training iterations and (c) CPINN + ACGD after $48\,000$ training iterations, and (d) the discriminator.

This problem has the manufactured solution

$$u(x, y) = \sin(x) \cos(y). \tag{16}$$

The PINN implementation has losses

$$\mathcal{L}_{\partial\Omega}^{\text{PINN}} = \frac{1}{N_{\partial\Omega}} \sum_{i=1}^{N_{\partial\Omega}} \left( \mathcal{P}(\overline{x}_i, \overline{y}_i) - u(\overline{x}_i, \overline{y}_i) \right)^2, \tag{17}$$

$$\mathcal{L}_{\Omega}^{\text{PINN}} = \frac{1}{N_{\Omega}} \sum_{i=1}^{N_{\Omega}} \left( \mathcal{P}_{xx}(x_i, y_i) + \mathcal{P}_{yy}(x_i, y_i) + 2 \sin(x_i) \cos(y_i) \right)^2, \tag{18}$$

and the CPINN losses are

$$\mathcal{L}_{\partial\Omega}^{\text{CPINN}} = \frac{1}{N_{\partial\Omega}} \sum_{i=1}^{N_{\partial\Omega}} \mathcal{D}_{\partial\Omega}(\overline{x}_i) \left( \mathcal{P}(\overline{x}_i, \overline{y}_i) - u(\overline{x}_i, \overline{y}_i) \right), \tag{19}$$

$$\mathcal{L}_{\Omega}^{\text{CPINN}} = \frac{1}{N_{\Omega}} \sum_{i=1}^{N_{\Omega}} \mathcal{D}_{\Omega}(x_i) \left( \mathcal{P}_{xx}(x_i, y_i) + \mathcal{P}_{yy}(x_i, y_i) + 2 \sin(x_i) \cos(y_i) \right). \tag{20}$$

Figure 2 (a) shows the exact solution of the PDE and the absolute error of the best models trained using (b) PINN and (c) CPINN, as well as an example of the discriminator output (d). The CPINN achieves lower errors than the PINN by a factor of about $10^6$ throughout most of the domain.

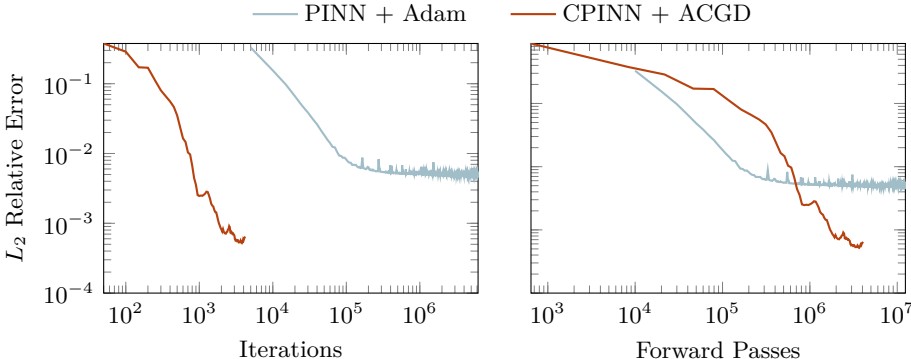

Figure 3: Comparison of CPINN and PINN on the nonlinear Schrödinger eq. (21) in terms of relative errors. After $200\,000$ training iterations, PINN cannot reduce the $L_2$ error further, plateauing about $4 \times 10^{-3}$, whereas CPINN reduces the error to $6 \times 10^{-4}$ under a smaller computational budget.

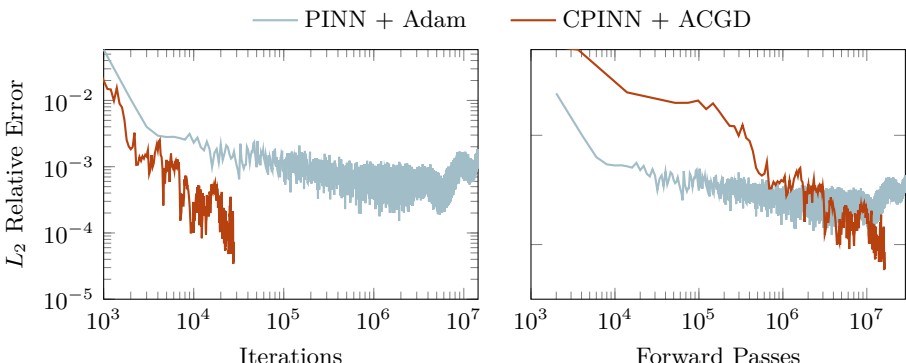

Figure 4: The relative errors of CPINNs (current) and PINNs on the Burgers' equation.

Figure 1 shows the relative $L_2$ error of a PINN and a CPINN on the Poisson problem. CPINN shows a faster convergence rate regarding the number of forward passes and training epochs, and its accuracy is higher than that of PINN by 4 orders of magnitude.

### 3.3 NONLINEAR SCHRÖDINGER EQUATION

In this subsection, we apply the competitive PINN methodology to solve the Schrödinger equation

$$u_t + \frac{1}{2}u_{xx} + |u|^2 u = 0, \quad x \in [-5, 5], \quad t \in [0, \pi/2], \tag{21}$$

where $u(t, x)$ is the complex-valued solution and

$$u(0, x) = 2\,\mathrm{sech}(x), \quad u(t, -5) = u(t, 5), \quad u_x(t, -5) = u_x(t, 5) \tag{22}$$

are the initial and boundary conditions.

The best results for CPINNs and PINNs are presented in fig. 3. CPINN reached an $L_2$ relative error of $6 \times 10^{-4}$ after $4.2 \times 10^3$ training iterations with $4.1 \times 10^6$ forward passes. PINN reached an $L_2$ relative error of $5 \times 10^{-3}$ after $6.2 \times 10^6$ training iterations, equivalent to $6.2 \times 10^6$ forward passes.

### 3.4 BURGERS' EQUATION

We next consider the case of the viscous Burgers' equation. This nonlinear second-order equation is

$$u_t + u u_x - (0.01/\pi)u_{xx} = 0, \quad x \in [-1, 1], \quad t \in [0, 0], \tag{23}$$

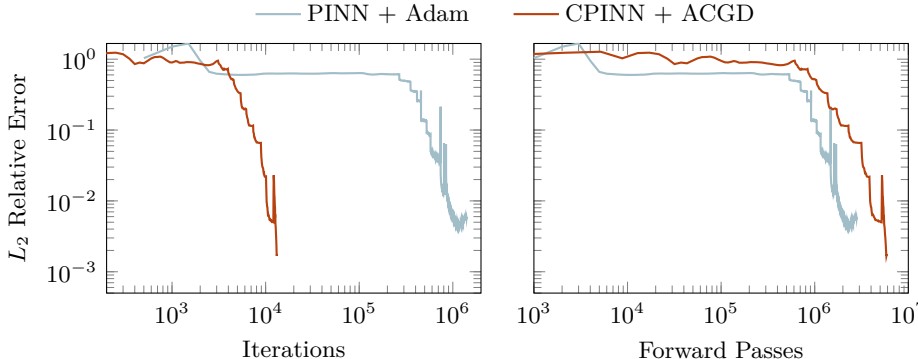

Figure 5: The relative error for the Allen–Cahn equation. A CPINN and a PINN that uses the *curriculum learning* approach of Wight & Zhao (2020) are shown. CPINN does not outperform this particular PINN, which may be due to the overall low accuracy of both methods.

which has parameters matching Raissi et al. (2019). $u(t, x)$ is the solution of the PDE and

$$u(0, x) = -\sin(\pi x), \quad u(t, -1) = u(t, 1) = 0 \tag{24}$$

are the initial and boundary conditions. A comparison to CPINNs is presented in fig. 4. CPINN exhibits an improved convergence rate and continues to reduce the error, whereas the progress of PINN eventually stagnates.

## 3.5 ALLEN–CAHN EQUATION

We next consider the one-dimensional Allen–Cahn equation with periodic boundary conditions, a cubically nonlinear equation given by

$$u_t - 0.0001u_{xx} + 5u^3 - 5u = 0, \quad x \in [-1, 1], \quad t \in [0, 1], \tag{25}$$

where $u(t, x)$ is the solution of the PDE and

$$u(0, x) = x^2 \cos(\pi x), \quad u(t, -1) = u(t, 1), \quad u_x(t, -1) = u_x(t, 1) \tag{26}$$

are the initial and boundary conditions. This example follows Raissi et al. (2019), and we use their training and testing data.

Raissi et al. (2019) does not report the performance of a standard PINN and instead shows the performance of a problem-specific modification that exploits the temporal structure of the problem. Wight & Zhao (2020) observe that this problem is difficult to solve directly for PINNs and propose a curriculum learning approach to remedy this problem.

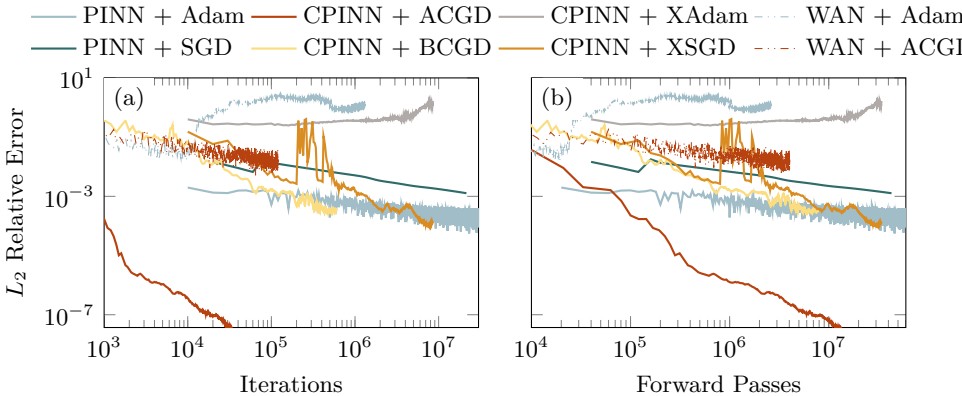

Figure 6: Comparison of PINNs and CPINNs with different optimizers for the Poisson problem. Following Zang et al. (2020), WAN + Adam uses Adagrad to update the max player.

Table 1: Performance of CPINNs and PINNs on the 2D Poisson problem of equation 15.

| | Optimizer | Iterations | $L_2$ Rel. Error | $\mathcal{L}^{\text{PINN}}$ | $\mathcal{L}^{\text{PINN}}_{\Omega}$ | $\mathcal{L}^{\text{PINN}}_{\partial\Omega}$ |
|---|---|---|---|---|---|---|
| PINN | Adam | $3 \times 10^{7}$ | $1.2 \times 10^{-4}$ | $1.4 \times 10^{-8}$ | $8.4 \times 10^{-8}$ | $5.5 \times 10^{-9}$ |
| | SGD | $2.2 \times 10^{7}$ | $1.3 \times 10^{-3}$ | $1.1 \times 10^{-6}$ | $5.2 \times 10^{-7}$ | $6.3 \times 10^{-7}$ |
| CPINN | ACGD | $4.8 \times 10^{4}$ | $1.7 \times 10^{-8}$ | $2.1 \times 10^{-14}$ | $2 \times 10^{-14}$ | $6 \times 10^{-16}$ |
| | CGD | $6 \times 10^{5}$ | $3.5 \times 10^{-4}$ | $5.1 \times 10^{-7}$ | $3.5 \times 10^{-7}$ | $1.7 \times 10^{-7}$ |
| | XAdam | $8.5 \times 10^{6}$ | $1.6$ | $1.6 \times 10^{6}$ | $1.6 \times 10^{6}$ | $0.2$ |
| | XSGD | $8.5 \times 10^{6}$ | $1.2 \times 10^{-4}$ | $1.2 \times 10^{-7}$ | $9.2 \times 10^{-8}$ | $3 \times 10^{-8}$ |
| WAN | Adam + Adagrad | $1.3 \times 10^{6}$ | $1.2$ | $2.6$ | $2.6$ | $1.2 \times 10^{-3}$ |
| | ACGD | $1.8 \times 10^{5}$ | $9 \times 10^{-3}$ | $1.1 \times 10^{-3}$ | $1.1 \times 10^{-3}$ | $2.7 \times 10^{-5}$ |

We use the approach of Wight & Zhao (2020) for PINNs and a suitable adaptation for CPINN. On this problem, PINNs modestly outperform CPINNs. This may be due to the slow convergence that prevents reaching the regime where PINNs begin plateauing. Using curriculum learning could improve this (Wight & Zhao, 2020).

## 3.6 OTHER OPTIMIZERS

We repeat our experiments with the Poisson problem using different optimizers to distinguish the effects of CPINN and ACGD. We use Adam and SGD for PINNs, and basic CGD (BCGD, without adaptive step sizes), ACGD, ExtraSGD (XSGD), and ExtraAdam (XAdam) for CPINNs (Korpelevich, 1977; Gidel et al., 2019). We also show results for Adam and ACGD on the weak adversarial network (WAN) of Zang et al. (2020). BCGD/ACGD applied to a single agent reduce to SGD/Adam. Thus, it is natural to compare CPINNs trained with BCGD/ACGD to PINNs trained with SGD/Adam.

We observe that combined with CPINN, the nonadaptive algorithms BCGD and extragradient improve over PINN trained with (the equally nonadaptive) SGD. The latter even improves over PINNs trained with Adam after sufficient iterations. We could not achieve training via ExtraAdam on the CPINN strategy. We were also unable to achieve high accuracy with WAN. The ACGD optimizer achieves much better results than all others. We conclude that CPINN improves over PINN and WAN. Realization of this advantage is enabled by the robust and adaptive training afforded by ACGD.

## 4 CONNECTIONS TO EXISTING WORK

Saddle point formulations arise in Petrov–Galerkin and mixed finite-element methods, which use different finite-dimensional function spaces to represent the solution (called trial functions) and measure the violation of the equations (called test functions) (Quarteroni & Valli, 2008; Fortin & Brezzi, 1991). The neural networks introduced here, $\mathcal{P}$ and $\mathcal{D}$, are nonlinear analogs of trial and test spaces. Unlike our approach, *Deep Petrov–Galerkin* of Shang et al. (2022) only parameterizes the trial space by a neural network and uses a conventional discretization such as finite elements for the test space. They only train the last layer while keeping all other layers fixed after initialization.

Saddle point problems in PDEs can be interpreted geometrically and have a game-theoretic interpretation (Lemaire, 1973). In a complementary game-theoretic approach, Owhadi (2017) cast computation as a game between a numerical analyst and an environment, using ideas from decision theory (Wald, 1945). Saddle point problems also arise from the introduction of Lagrange multipliers, which are used to cast a constrained optimization problem as an unconstrained saddle point problem (Brezzi, 1974). These discriminators can be viewed as neural-network-parametrized Lagrange multipliers that enforce distributional equality (in GANs) or satisfaction of the PDE (in CPINNs).

Our work is thus related to recent efforts that combine CGD with Lagrange multipliers to solve constrained optimization problems arising from reinforcement learning and computer graphics (Bacon et al., 2019; Yu et al., 2021; Soliman et al., 2021). The need for constrained training of neural networks also arises in other applications, resulting in an increasing body of work on this topic (Pathak et al., 2015; Donti et al., 2021; Lokhande et al., 2020; Fioretto et al., 2020).

Following section 1, *Deep Ritz* exploits the fact that some PDEs can be cast as minimization problems without squaring the condition number (E & Yu, 2017), expressing equation 11 as

$$\min_{\boldsymbol{\pi}} \frac{\boldsymbol{\pi}^\top \boldsymbol{A} \boldsymbol{\pi}}{2} - \boldsymbol{\pi}^\top \boldsymbol{f},$$

for a symmetric positive definite matrix $\boldsymbol{A}$. Such a formulation is not always available, and even if it exists, one still must enforce the PDE boundary conditions (Liao & Ming, 2019). Zang et al. (2020) uses a weak PDE formulation to derive a minimax formulation. In the notation of section 2.3, their approach is similar to using the zero-sum game

$$\min_{\boldsymbol{\pi}} \max_{\boldsymbol{\delta}} \log \left( \left( \boldsymbol{\pi}^\top \boldsymbol{A} \boldsymbol{\delta} \right)^2 \right) - \log \left( \|\boldsymbol{\delta}\|^2 \right).$$

In contrast to our work, this formulation encourages the max-player only to learn *where* the min-player violates the PDE, not *how* (the sign of the residual) the PDE is being violated.

Many works adapt the penalty weights in PINNs during training to improve accuracy, often using minimax formulations (Wang et al., 2021a; Xu et al., 2021; McClenny & Braga-Neto, 2020; van der Meer et al., 2022). Different from the $\mathcal{D}$ output of CPINNs, these weights are multiplied with the *square* violation of the equality constraint, which is always greater than zero. Therefore, they do not correspond to a meaningful zero-sum game because the optimal discriminator strategy, in this case, drives all weights to infinity. Alternatively, Krishnapriyan et al. (2021) recommends the use of curriculum learning. They train the initial PINN on a better-conditioned variant of the original problem that slowly transforms into the target problem during training.

The discussion in section 2.3 seemingly contradicts the works of Soliman et al. (2021); Lu et al. (2022) on Sobolev acceleration, whereby training on higher order derivatives of the loss can lead to improved accuracy. As discussed in appendix A.7 of the appendix, we believe that Sobolev acceleration can be reconciled with the arguments presented in section 2.3. However, a rigorous understanding of the relationship of these two perspectives is presently missing and will require additional research.

## 5 DISCUSSION

This work introduced CPINNs, an agent-based approach to the neural-network-based solution of PDEs. CPINNs are crafted to avoid the ill-conditioning resulting from traditional PINNs least squares loss functions. Section 3 showed that CPINNs trained with ACGD improve upon the accuracy of PINNs trained with a comparable computational budget and can solve PDEs beyond even single-precision floating-point accuracy, the first approach with this capability.

With CPINNs, one can now achieve single-precision errors, but significant computation is still required. This is due to the number of CG iterations performed for each ACGD step. Reducing this overhead is a direction for future work. Potential solutions to this problem include cheaper approximate solutions of the matrix inverse, different iterative solvers, or the recycling of Krylov subspaces (Paige & Saunders, 1975; Parks et al., 2006). It may also be worth investigating alternatives to ACGD with lower per-iteration cost, such as the methods proposed by Tan et al. (2018).

The experiments used the same training points for each iteration, following Raissi et al. (2019). In the future, we will investigate the effects of batch stochasticity on training accuracy. We also plan to investigate competitive mirror descent (CMD) for partial differential inequalities and contact problems (Schäfer et al., 2020a; Lions, 1972) and, more generally, CPINN-like approaches to other problems involving the constrained training of neural networks.

### ACKNOWLEDGMENTS

This research was supported in part through research cyberinfrastructure resources and services provided by the Partnership for an Advanced Computing Environment (PACE) at the Georgia Institute of Technology, Atlanta, Georgia, USA. SHB acknowledges the NVIDIA Academic Hardware Grant Program for facilitating part of this work. We thank Hongkai Zheng for maintaining the CGDs package used to conduct this work.

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

# A APPENDIX

This appendix details the experiments mentioned in Section 3 and additional plots for the solutions. We also added a visualization of the CPINN method for a more clear understanding

All PINN models and CPINN generators use the hyperbolic tangent as the activation function. Discriminators in CPINNs contain ReLU as the activation function. The number of neurons and layers in the neural networks varies for each equation. We use the GMRES-based ACGD implementation of Zheng (2020), available under the MIT license at `https://github.com/devzhk/cgds-package`, as well as the implementation of ExtraGradient methods available under the MIT license at `https://github.com/GauthierGidel/Variational-Inequality-GAN`. For the experiments in section 3.3, 3.4, 3.5 we use the training and testing data sets from `https://github.com/maziarraissi/PINNs/tree/master/main`, which are available under the MIT license. For the Weak Adversarial Network experiments we used code from `https://github.com/yaohua32/wan`, available under the MIT lience.

## A.1 VISUALIZATION

In fig. 7 we provide a visualization of PINN and CPINN on an example of Poisson problem

$$\Delta u(x) = f \quad \text{in the domain } \Omega \tag{27}$$
$$u(x) = g \quad \text{on the boundary } \delta\Omega. \tag{28}$$

In PINN, we feed training set $x$ to PINN with weights $\sigma$. PINN output $u$ and we can calculate $\Delta u$ with respect to the input. The loss function $\mathcal{L}^{\text{PINN}}$ of PINN is equal to the sum of the mean square residual between $g$ and $u$ on the boundary and the mean square residual between $f$ and $u_{bc}$ in the domain.

For CPINN, similarly, we feed training set $x$ to PINN with weights $\sigma$. PINN outputs $u$, and we can calculate $\Delta u$ with respective to the input. We also feed the training set $x$ to a discriminator network with weights $\theta$. The discriminator predicts the residuals associated with PINN and outputs two vectors of bet coefficients, $d_{bc}$ and $d_{\Delta u}$, to penalize the domain residual associated with eq. (27) and the boundary value residual eq. (28). The loss function $\mathcal{L}^{\text{CPINN}}$ of CPINN is equal to the sum of the product of discriminator bet vectors and residuals of PINN.

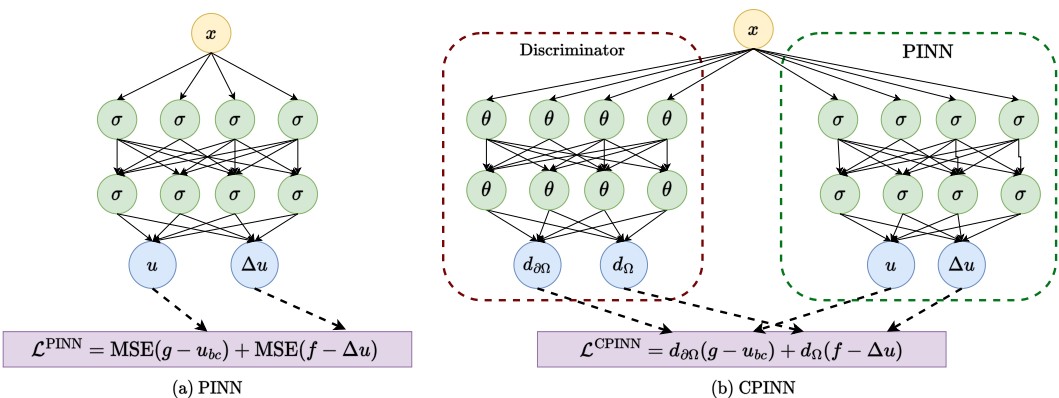

(a) PINN      (b) CPINN

Figure 7: Illustration of (a) PINN and (b) CPINN.

## A.2 POISSON EQUATION

For the Poisson equation, we use $5\,000$ training points within the domain $[-2, 2] \times [-2, 2]$, $50$ training points on each side of the domain boundary. We randomly selected all training points with Latin Hypercube sampling. The PINN model contains 3 hidden layers with $50$ neurons in each layer. The discriminator in the CPINN contains 4 hidden layers with $50$ neurons in each layer.

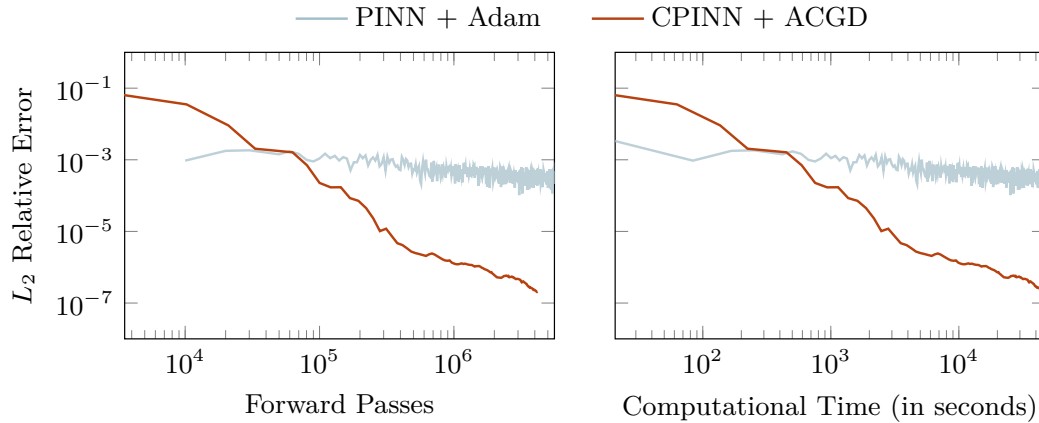

Figure 8: Comparison of PINN and CPINN in terms of the number of forward passes and computational time. Note that the two plots appear almost identical, establishing that forward passes are an implementation-independent computational cost proxy.

The PINNs are trained with Adam (Kingma & Ba, 2014) and SGD (Ruder, 2016). Adam and ACGD both use a learning rate of $10^{-3}$, beta values $\beta_1 = 0.99$ and $\beta_2 = 0.99$. The $\epsilon$ of Adam and ACGD are each set to $10^{-8}$ and $10^{-6}$, respectively (all parameters following the usual naming conventions).

We also included a comparison of PINN and CPINN in fig. 8 in terms of both the forward passes and computational time to compare the computational cost. We train both models on an NVIDIA V100 GPU.

### A.3 SCHRÖDINGER'S EQUATION

We use Latin Hypercube sampling to randomly select 20 000 training points within the domain and 50 points on each boundary.

We test several different network configurations and hyperparameters. This includes combinations of the number of neurons per layer for each network, including 100 and 200 neurons per layer; learning rates of $10^{-5}, 10^{-4}, 5 \times 10^{-4}, 10^{-3}, 10^{-2}$, and $2 \times 10^{-2}$; and Adam and ACGD $\beta$ values of $(0.99, 0.99)$ and $(0.9, 0.999)$.

We chose $10^{-4}$ and $10^{-3}$ as the learning rate, $(0.9, 0.999)$ and $(0.99, 0.99)$ as the $\beta$ values, $10^{-8}$ and $10^{-6}$ as the $\epsilon$ for Adam and ACGD, respectively. The iterative linear solve of the ACGD optimizer uses a relative tolerance of $10^{-7}$ and absolute tolerance of $10^{-20}$.

The PINN presented in fig. 3 contains 4 hidden layers with 100 neurons per layer, and the discriminator in CPINN contains 4 hidden layers with 200 neurons per layer.

Figure 9 (a) shows the exact solution of the PDE and the absolute error of the best models trained using (b) PINN and (c) CPINN, as well as an example of the discriminator output (d).

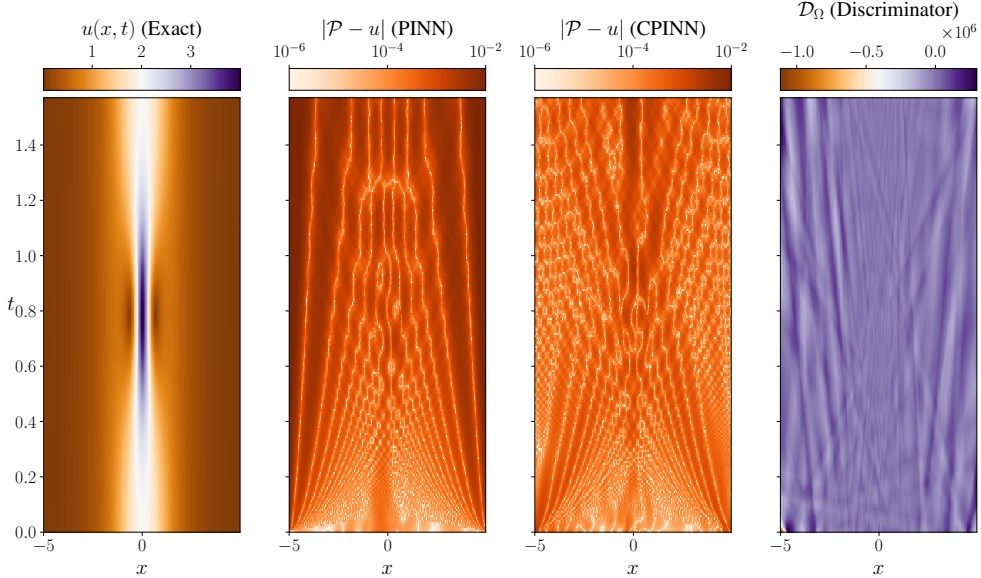

Figure 9: (a) Exact solution $u$ to eq. (21), absolute errors of (b) PINN + Adam after $6.2 \times 10^6$ training iterations and (c) CPINN + ACGD after $4\,200$ training iterations, and (d) the discriminator.

### A.4 BURGERS' EQUATION

For the Burgers' equation, we use Latin Hypercube sampling to randomly select $10\,000$ training points within the domain, 100 points at the initial condition and 100 points for each boundary condition specified in eq. (24).

The PINN model contains 8 hidden layers with 60 neurons per layer, and the discriminator in the CPINN contains 8 hidden layers with 60 neurons per layer.

The $\beta$ of Adam and ACGD are $(0.99, 0.99)$, and the learning rates of both optimizers are $10^{-3}$. The $\epsilon$ are $10^{-8}$ and $10^{-6}$ for Adam and ACGD, respectively. The iterative linear solve of the ACGD optimizer had a relative tolerance of $10^{-7}$ and an absolute tolerance of $10^{-20}$.

Figure 10 (a) shows the exact solution of the PDE and the absolute error of the best models trained using (b) PINN and (c) CPINN, as well as an example of the discriminator output (d).

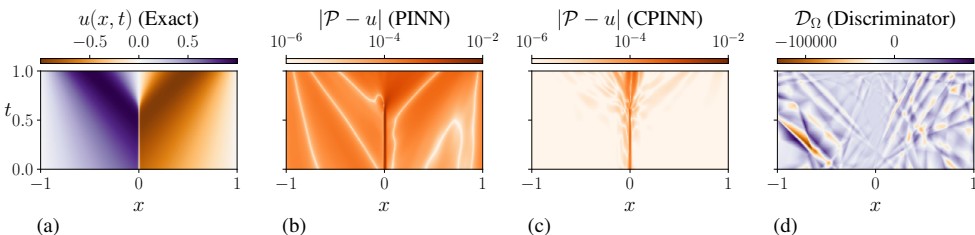

Figure 10: (a) Exact solution $u$ to eq. (23), absolute errors of (b) PINN + Adam after $1.4 \times 10^6$ training iterations and (c) CPINN + ACGD after $28\,000$ training iterations, and (d) the discriminator.

### A.5 ALLEN-CAHN EQUATION

For the Allen Cahn equation, we randomly select $10\,000$ training points within the domain for the PDE constraint in eq. (25), 100 and 256 points for the initial condition and boundary condition

specified in eq. (26), respectively. The learning rates of Adam and ACGD are both $10^{-3}$, the $\beta$ of Adam and ACGD are $(0.99, 0.99)$, and the $\epsilon$ are $10^{-8}$, $10^{-6}$, respectively.

The PINN model contains 4 hidden layers with 128 neurons per layer for the adaptive sampling method experiments. The discriminator in the CPINN contains 4 hidden layers with 256 neurons per layer. GMRES was used as the inner iterative solver in ACGD, with a relative tolerance of $10^{-7}$ and an absolute tolerance of $10^{-20}$.

For the curriculum learning approach, we divide the $10\,000$ points into 10 subsets based on the time coordinate. Once the $\mathcal{L}_{\Omega}^{\mathrm{PINN}}$ on the current training set is less than $10^{-7}$, we include the next subset of collocation points in the training.

Figure 11 (a) shows the exact solution of the PDE and the absolute error of the best models trained using (b) PINN and (c) CPINN, as well as an example of the discriminator output (d).

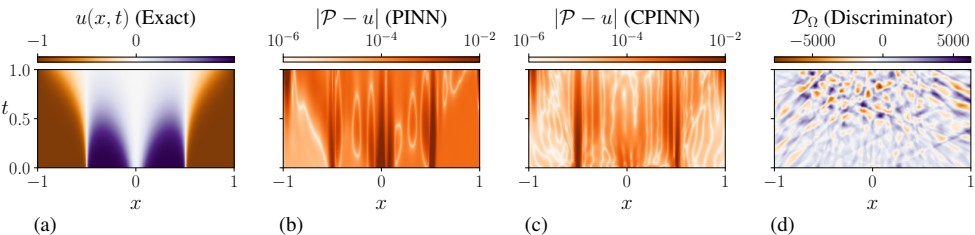

(a)    (b)    (c)    (d)

Figure 11: (a) Exact solution $u$ to eq. (25), absolute errors of (b) PINN + Adam after $1.4 \times 10^{6}$ training iterations and (c) CPINN + ACGD after $13\,250$ training iterations, and (d) the discriminator.

### A.6    OTHER OPTIMIZERS

In addition to ACGD and Adam, we also use SGD, CGD (Schäfer & Anandkumar, 2019), ExtraAdam and ExtraSGD (Gidel et al., 2019) to approximate the solution of equation 6 in section 3.6. We test several hyperparameter combinations for each optimizer. The learning rates vary included $10^{-4}, 5 \times 10^{-4}, 10^{-3}, 10^{-2}$, and $2 \times 10^{-2}$. The $\beta$ of Adam and ACGD are $(0.99, 0.99)$. We tested several $\beta$ values for Adam and ACGD but did not observe meaningful changes to our results or conclusions when varying them. For the ExtraAdam optimizer, we tested several pairs of $\beta$ values, from $0.3, 0.5, 0.7, 0.9, 0.99$, though none provide accuracy competitive with the other optimizers. For CGD, SGD and ExtraSGD, the learning rates are $10^{-2}$. The CGD optimizer uses conjugate gradient as the iterative solver with relative tolerance $10^{-12}$ and absolute tolerance $10^{-20}$.

We tried several different setups for the Weak Adversarial Network experiments, including using a fixed set of training points or randomizing the training set per epoch, removing or preserving the logarithm in PINN loss, and different combinations of activation functions. We present the results of the best models optimized with either GACGD or the combination of Adam and AdaGrad in fig. 6.

In one WAN experiment we used Adam for the optimization of PINN with a learning rate of $10^{-3}$ and $(0.9, 0.999)$ as $\beta$ values, AdaGrad for the optimization of the discriminator, with a learning rate of $1.5 \times 10^{-3}$. We resample $5\,000$ interior points in the domain and 200 boundary points per training epoch. The PINN model contains 6 hidden layers with 40 neurons per layer, with softplus as the activation functions for layers 1, 2, and 4, sin as the activation function for layers 3 and 5, and no activation in the last layer. The discriminator, or network $\phi$, contains 8 hidden layers with 40 neurons per layer. The activation functions are set to $\tanh$ for layers 1, softplus for layers 2, 4, and 6, sin for layers 3, 5, and 7, and no activation in the last layer. All model and optimizer settings in the first experiment are identical to the setup used in section 4.2.2 of Zang et al. (2020). We use the training procedure described in algorithm 3 of Zang et al. (2020) and removed the logarithm operation in the loss of PINN as mentioned in section 3.5 of Zang et al. (2020).

In the second WAN experiment, we used ACGD for the optimization, with a learning rate of $10^{-3}$ and $\beta$ of $(0.99, 0.99)$. The iterative linear solve of the ACGD optimizer had a relative tolerance of $10^{-7}$ and an absolute tolerance of $10^{-20}$. We used a fixed training set with $5\,000$ interior points in

the domain and 200 boundary points during the training. The PINN model contains 8 hidden layers with 40 neurons per layer, with softplus as the activation functions for layers 1, 2, 4 and 6, $\sin$ as the activation function for layers 3, 5 and 7, and no activation in the last layer. The discriminator, or network $\phi$, contains 8 hidden layers with 40 neurons per layer. The activation functions are set to $\tanh$ for layers 1, softplus for layers 2, 4, and 6, $\sin$ for layers 3, 5, and 7, and no activation in the last layer. We use the training procedure as described by algorithm 1 of Zang et al. (2020) and removed the logarithm operation in the loss of PINN as mentioned in section 3.5 of Zang et al. (2020).

## A.7 RELATIONSHIP WITH SOBOLEV ACCELERATION

We believe that the improvements due to Sobolev training observed by Son et al. (2021), as well as the theoretical analysis by Lu et al. (2022), can be reconciled with the perspective of section 2.3.

Lu et al. (2022) model PINN-training as a kernel method with a kernel that is jointly diagonalizable with the differential operator $\mathcal{A}$, such that the former has polynomially increasing eigenvalues while the latter has polynomially decreasing eigenvalues. Thus, the kernel matrix and the associated feature map $\mathcal{S}$ behave like a power of the inverse of the differential operator $\mathcal{A}$. After cancellation, the Hessian $\mathcal{S}^*\mathcal{A}^*\mathcal{A}\mathcal{S}$ behaves either like a power of $\mathcal{A}$ or a power of its inverse. The reasoning of Lu et al. (2022) is that the additional differential operators due to Sobolev training cancel with excess powers of $\mathcal{A}^{-1}$, thus improving the conditioning. For instance, Sobolev training may improve the conditioning by replacing the Hessian $\mathcal{S}^*\mathcal{A}^*\mathcal{A}\mathcal{S}$ with $\mathcal{S}^*\mathcal{A}^*\mathcal{A}^2\mathcal{S}$. In contrast, CPINNs replace the Hessian $\mathcal{S}^*\mathcal{A}^*\mathcal{A}\mathcal{S}$ with $\mathcal{R}^*\mathcal{A}\mathcal{S}$, where $\mathcal{R}$ is the feature map of the discriminator network. Under the assumption that $\mathcal{R} = \mathcal{S} = \mathcal{A}^{-2/3}$, Sobolev training leads to a well-conditioned system while CPINNs leads to a worse condition number than PINNs. However, $\mathcal{A}^{-1} \approx \mathcal{R} \approx \mathcal{S}$ is a strong assumption that is unlikely to hold exactly. Absent this assumption, or if it only holds approximately, $\mathcal{A}\mathcal{S}$ remains ill-conditioned in a way that can not be fully mitigated by multiplying with powers of $\mathcal{A}$. While $\mathcal{R}$ may also be ill-conditioned, its large and small eigenvalues are not perfectly aligned with those of $\mathcal{A}\mathcal{S}$, leading to possibly significantly better conditioning of $\mathcal{R}^*\mathcal{A}\mathcal{S}$ compared to $S^*\mathcal{A}\mathcal{S}$. For instance, if a single large eigenvalue of $\mathcal{A}$ is not matched by a small eigenvalue of $\mathcal{S}$, it will appear squared in $\mathcal{S}^*\mathcal{A}^*\mathcal{A}\mathcal{S}$. However, it could be attenuated by a small eigenvalue of $\mathcal{R}$ in $\mathcal{R}^*\mathcal{A}\mathcal{S}$.

Sobolev training furthermore assumes regularity of the PDE residual. Thus, it may bias the network to more regular solutions, thus improving generalization in sufficiently regular problems. This may explain the improved test errors observed by Son et al. (2021) and Lu et al. (2022).

