# OpenReview forum: "Competitive Physics Informed Networks "
_ICLR.cc/2023/Conference — ICLR 2023 poster_

### Official Review · Reviewer_QL5E · 2022-10-14

**Confidence:** 5
**Clarity, Quality, Novelty And Reproducibility:** The writing of this paper is super gr…
**Correctness:** 3
**Technical Novelty And Significance:** 3
**Empirical Novelty And Significance:** 4
**Recommendation:** 8

**Strength And Weaknesses:**

Strength: The experiments look strong, and the writing is super clear to understand.

Weakness: Several claims are hand waving, and the reviewer is not convinced. I think if the paper is accepted at this stage, it will make confusion in the community.

**Regards Experiment** The reviewer's first major concern is the relationship of this paper with [1] which also uses the weak formulation to train a PDE solution.  I think an experiment comparing these works is essential (using the same optimization algorithm).

The paper only includes results comparison with respect to the number of iterations. Comparison with respect to the computation time is also needed. For CPINN introduces a new NN and CGD have a matrix inverse. Further computation cost have been included.

**Regards Mni-max** min-max optimization is hard. The complexity lower bound proposed of minimax optimization has a term K_x K_y [2]. I guess using min-max has the same computational complexity as the original formulation.

To reduce the condition number of laplace squares, we can still use PINN to do this by considering
min_{u_1,u} ||u_1-\nabla u||^2+||f-\nabla dot u_1||^2
to solve Delta u = f. I don't think the minimax formulation is essential.

At the same time, the paper uses ADAM, which is an adaptive method. The adaptive methods can cancel the condition number. I'm confused by this mismatch of the experiment with the theory claimed.

**Regards the Squared residuals.** Changing the loss into the Sobolev norm also introduces a further matrix A. But [3,4] showed this would only accelerate the training both in experiments and theory for the machine learning algorithms. This paper have considered the theory as [4] includes a machine learning model into the learning objective. Can the authors comment on this?

**Related work [5]** The reviewer thinks the paper is still valuable. [5] proposed a similar trick from the optimization community and figured out the improvement of this method.  I encourage the authors to think about combining the theory in [4] and [5] to figure out the true reason this method has an acceleration.

[1] Yaohua Zang, Gang Bao, Xiaojing Ye, and Haomin Zhou. Weak adversarial networks for high-dimensional partial differential equations. Journal of Computational Physics, 411:109409, 2020

[2] Zhang J, Hong M, Zhang S. On lower iteration complexity bounds for the convex-concave saddle point problems. Mathematical Programming, 2022, 194(1): 901-935.

[3] Son H, Jang J W, Han W J, et al. Sobolev training for the neural network solutions of pdes[J]. arXiv preprint arXiv:2101.08932, 2021.

[4] Lu Y, Blanchet J, Ying L. Sobolev Acceleration and Statistical Optimality for Learning Elliptic Equations via Gradient Descent[J]. arXiv preprint arXiv:2205.07331, 2022.

[5] https://proceedings.neurips.cc/paper/2018/file/08048a9c5630ccb67789a198f35d30ec-Paper.pdf


===

The author's response convinced me this is a worth pursuing direction, I've raised my score.


**Summary Of The Paper:**

This paper introduce competitive physics-informed networks where two neural networks solve a partial differential equation by playing a zero-sum game. The formulation is built using a weak form. The authors claim that this weak form have smaller condition number.

**Summary Of The Review:**

The experiments look strong, and the writing is super clear to understand. Several claims are hand waving, and the reviewer is not convinced. I think if the paper is accepted at this stage, it will make confusion in the community.

I still consider this method have made a huge contribution to the PINN family. I suggest authors to investigate paper [5] in detail to see the benefit of this method.

---

> ### Author Response · Authors · 2022-11-15
> **Response to reviewer QL5E, Part 2/2**
>
> ## Sobolev training:
> We agree that our work is seemingly at odds with the idea of Sobolev training, in particular the analysis by Lu et al., according to which penalizing Sobolev norms of the PDE residual can improve PINN training performance.
> We reconcile these two works as follows (we also have added this discussion to the appendix of the revised version.
>
> ### Spectral cancellation:
> Lu et al. model PINN training as a kernel method with a kernel that is jointly diagonalizable with the differential operator such that the former has polynomially increasing eigenvalues while the latter has polynomially decreasing eigenvalues (Assumption 1 in Lu et al.).
> Thus, the kernel matrix and the associated feature map $S$ behave like a power of the inverse of the differential operator $A$.
> After cancellation, the Hessian $S^* A^* A S$ behaves either like a power of $A$ or a power of its inverse.
> The reasoning of Lu et al. is that the additional differential operators due to Sobolev training cancel with excess powers of $A^{-1}$, thus improving the conditioning. For instance, Sobolev training may improve the conditioning by replacing the Hessian $S^* A^*  A S \rightarrow S^* A^* A^2 S$. In contrast, CPINNs replace the Hessian $S^* A^*  A S \rightarrow R^* A S$, where $R$ is the feature map of the discriminator network. Under the assumption that $R = S = A^{-2/3}$, Sobolev training leads to a well-conditioned system while CPINNs lead to a worse condition number than PINNs. However, $A^{-1} \approx R \approx S$ is a strong assumption that, to the best of our knowledge, is not supported by empirical evidence. Absent this assumption, $AS$ remains ill-conditioned in a way that can not be fully mitigated by multiplying with powers of $A$.  While $R$ may also be ill-conditioned, its large and small eigenvalues are not perfectly aligned with those of $AS$, leading to possibly significantly better conditioning of $R^* A S$ than $S^* A S$. For instance, if a single large Eigenvalue of $A$ is not matched by a small eigenvalue of $S$, it is going to appear squared in $S^* A^*  A S$, whereas it may be attenuated by a small eigenvalue of $R$ in $R^* A S$.
>
> ### Smoothness assumptions:
> We point out that the use of Sobolev training implicitly assumes regularity of the PDE residual. This has two important implications.
> First, it means that Sobolev training relies on higher-order regularity, precluding its application to problems with rough coefficients.
> Second, it makes it possible for Sobolev training to improve the generalization performance by guiding the network to rely on higher-order regularity. The works of both Son et al. and Lu et al. only provide testing loss/error, making it difficult to rule out this explanation.
>
> ## Related work and theory:
> Thank you for sharing this work. Since ACGD outperformed the extragradient method in our experiments (even after accounting for the cost of the matrix inverse), we did not pursue $\mathcal{O}(1)$ methods further. However, we believe that there is still ample space for improving the optimization of CPINNs. To this end, the method by Tan et al. is a promising direction.
> We also thank the reviewer for their pointers regarding theoretical approaches. Section 2.3 presents our motivation for introducing CPINNs and its relationship to conventional wisdom in numerical PDEs. However, we agree that a more thorough understanding would be highly desirable. We see this as an exciting direction for future work. We have added a sentence to the discussion that points out this possibility.

---

> > ### Comment · Reviewer_QL5E · 2022-11-15
> > **Most of my concerns are addressed but…**
> >
> > Most of my concerns are addressed. Thanks the author for the hard work for doing the rebuttal. However, I still have the following two concerns before I change the score.
> >
> > The statement in Chapter 3 in (Greenbaum , 1997) that the minimax formulation allow for a significantly (both in practice and asymptotically as a function of the condition number) faster solution than what would be possible by matrix-vector products is interesting to me. Can the author show how the PDEs you are considering fits in this framework.
> >
> > The second is about the evaluation in forward passes. I think evaluation in computational time is still important. For the backward process of matrix vector product and bp for gradient are different. I would see the comparison even just put in the appendix. There is still time before the discussion deadline.

---

> > > ### Author Response · Authors · 2022-11-16
> > > **Thank you for your quick response! Part 2/2**
> > >
> > > ### Cost of Hessian-vector products
> > > You are right that computing the gradient of $x \mapsto \nabla_{y} f(x, y) \cdot v$ may indeed be more expensive than computing the gradient of $\nabla_{y} f(x, y)$ and the former is indeed how the matrix-vector product $[D_{xy}^2 f] v$ is presently implemented.
> > > However, we want to point out the following.
> > >
> > > 1. We know that computing a gradient is not substantially more expensive than evaluating the function itself. Computing the gradient of $\nabla_{y} f(x, y) \cdot v$ is therefore not substantially more expensive than computing the gradient $\nabla_{y}f(x,y)$, and thus computing $f(x, y)$.
> > >
> > > 2. The use of double-backprop, also known as "reverse over reverse" for computing the Hessian-vector products is not necessary. Instead, one could use "forward over reverse", which can be implemented using dual numbers. With this approach, the additional cost should be comparable to the difference between using Real and Complex floating point numbers. For instance, the [JAX manual](https://jax.readthedocs.io/en/latest/notebooks/autodiff_cookbook.html) reports a 2.5-fold speed-up when using forward over reverse AD instead of reverse over reverse.
> > >
> > > 3. In principle, the non-infinitesimal part of the dual number computation is already performed when taking the gradient. Thus, only the infinitesimal part of the dual numbers has to be recomputed for each Hessian-Vector product. This should have almost exactly the same computational cost as computing a single gradient. This is the infinitesimal analog of computing the Hessian vector product using finite differences, at the cost of a single additional gradient computation.
> > >
> > > That being said, we understand and agree with your perspective that having concrete wall-clock times would be helpful. The experiments in the paper were run on different computational hardware and shared nodes under varying loads, so even if we had kept the wall-clock times, they would not be a reliable measure of performance. To remedy this issue, we have rerun the Poisson experiment, logging the wall-clock times. The resulting accuracy-vs-time plot is contrasted with the accuracy-vs-forward passes plot in Figure 8 in the Appendix of the revised version. Note that the relative convergence speed of CPINN and PINN is almost indistinguishable, establishing forward passes as a faithful proxy for computational cost.
> > >
> > > We want to thank you again for your thoughtful review and your responsiveness during the discussion. Please let us know if you have any further questions.

---

> > > ### Author Response · Authors · 2022-11-16
> > > **Thank you for your quick response! Part 1/2**
> > >
> > > Thank you for your quick response! Please see below for our comments regarding your remaining two concerns.
> > >
> > > ### Improvements due to minmax
> > > Equation 3.12 of (Greenbaum, 1997) states that for a positive-definite matrix with a given condition number bound $\kappa$, $k$ matrix-vector products can reduce the residual by a factor of $2 \frac{\left\|r_k\right\|}{\left\|r_0\right\|} \leq \left(\frac{\sqrt{\kappa} - 1}{\sqrt{\kappa} + 1}\right)^k$. The one-dimensional finite difference discretization of the Laplace operator with Dirichlet boundary conditions provides a matching lower bound. An algorithm that only has access to matrix-vector products with $A^{\top} A$ will therefore, in the worst case, be beholden to the above convergence rate with $\kappa = \left(\kappa(A)\right)^2$. Gradients of the squared residual $\|A \pi - f\|^2$ implement the map $\pi \mapsto \left(A^{\top} A \pi - A^{\top} f \right) / 2$. As a result, any method based on gradient descent applied to the squared residual will be restricted to the above bound with the squared condition number.
> > > In contrast, gradients of the objective of the minmax formulation can be used to implement matrix-vector products with both $A$ and $A^{\top}$. Therefore, they are not beholden to the squared condition number bounds.
> > >
> > > For instance, if the matrix were symmetric positive-definite, we could use matrix vector products with $A$ to converge according to the above bound with $\kappa = \kappa(A)$. Since the (negative) Laplace operator is self-adjoint and positive with respect to $L^2$ inner product, standard Galerkin discretizations of it are equally symmetric-positive definite (at least if the two sets of basis functions $\psi$ and $\phi$ are identical). As a result, such a discretized Poisson problem could be solved significantly faster from access to gradients of the minmax formulation instead of gradients of the squared residual. Many PDE's are not self-adjoint positive definite with respect to the $L^2$-inner product. However, their leading-order-term (the one with the highest order of derivatives) is often an elliptic operator such as the Laplacian, that has this property. As a result, the largest eigenvalues of the operator may still behave like that of an elliptic PDE, allowing for computational savings due to access to matvecs with $A$.  For instance, a discretized Helmholtz equation has the form $(A - \alpha I)\pi = f$, where $A$ is a discretized negative Laplacian. Let $\lambda_{0}$ be the smallest (in magnitude) eigenvalue of $(A - \alpha I)$, $\lambda_{\max}$ its most positive, and $\lambda_{\min} \approx -\alpha$ its most negative eigenvalue. In this example, matrix-vector products with $A^{\top}A$ would need to use the condition number bound $\kappa = \left|\lambda_{\max} / \lambda_{0} \right|^2$. By Equation 3.14 in Greenbaum, access to matvecs with $(A - \alpha I)$ would instead allow using $\kappa = \left|\lambda_{\max} / \lambda_{0}\right| \left| \alpha / \lambda_{0}\right|$. In the limit of a fine discretization we have $\lambda_{\max} \gg \alpha$, leading to substantial improvements.
> > >
> > > The above is meant to give some examples of systems where access to gradients of the minimax objective allows, in principle, a substantially more efficient solution of the problem than access to gradients of the squared residual.
> > > Of course, as soon as neural networks are involved, it is hard to make any precise mathematical statements.
> > > However, since matvecs with $A$ and $A^{\top}$ can always be used to compute a matvec with $A^{\top} A$, but not the other way around, it seems plausible that minimax-based approaches can do at least as well, and sometimes better, than methods based on minimizing the squared residual.

---

> ### Author Response · Authors · 2022-11-15
> **Response to reviewer QL5E, Part 1/2**
>
> Thank you for your careful review of our work. You made a number of insightful comments that helped us improve the paper. We respond to them below, in the order in which they were raised.
>
> ## Regarding experiments:
> ### Comparison to WAN:
> We have results for weak adversarial networks (WAN), trained with Adam or ACGD, to Figure 6. We observe that ACGD improves the stability of WAN training compared to the combination of Adam and Adagrad used by Zang et al. However, we were unable to achieve accuracy compared to CPINN with ACGD despite trying different sets of activation functions, loss functions (dropping the logarithm as suggested in section 3.5 of Zang et al.), and training modes (minibatching vs fixed batch). In summary, we believe that Figure 6 now makes a compelling case that the improvements of CPINN are not just a function of the optimizer used for its training.
>
> ### Computational Cost:
> We believe that we have already accounted for the additional costs associated with the CPINN methodology. This was done by tracking the number of ``forward passes,'' which accounts for the gradient or Jacobian-vector product computation with respect to either network, including those used by the iterative solution of the matrix-inverse in the update of rule of (A)CGD. These are the only additional components that make CPINN potentially more (or less) costly than a standard PINN. We use this measure of computational cost since it is independent of hardware and implementation.
>
>
>
>
>
>
> ## Regarding min-max:
> ### Complexity lower bounds:
> We believe that you are correct that the asymptotic complexity for a generic matrix $\mathbf{A}$, with difficulty measured by the condition number of $\mathbf{A}$, can not be improved by simply switching to min-max formulation. In particular, directly applying a Krylov subspace method to the symmetric dilation $\left(\begin{smallmatrix} 0 & \mathbf{A}^{\top} \\\\ \mathbf{A} & 0\end{smallmatrix}\right)$ is not going to be any more efficient than applying a Krylov method to $\mathbf{A}^{\top} \mathbf{A}$. However, there exist many matrices, including those that are symmetric positive-definite or those that are symmetric with a different condition number on the positive and negative real line, where matrix-vector products with $\mathbf{A}$ allow for a significantly (both in practice and asymptotically as a function of the condition number) faster solution than what would be possible by matrix-vector products with $\mathbf{A}^{\top}\mathbf{A}$, see Chapter 3 in [(Greenbaum , 1997)](https://epubs.siam.org/doi/book/10.1137/1.9781611970937). We have modified Section 2.3 to make these caveats more clear.
>
> ### Alternative way of reducing the condition number:
> The approach that you suggest could indeed help to reduce the condition number arising from the partial differential operator, since the resulting gradient can implement multiplication with the PDE operator instead of its square. Similar approaches are commonly used in the solution of fourth-order equations, transforming them into a system of second-order equations. The caveat of this approach is that it requires the a priori knowledge of a factorization of the ill-conditioned operator (which is available for differential operators). It is, therefore, unable to address the ill-conditioning that arises in the network Jacobian. Beyond this caveat, the practical improvements of this approach would need to be verified empirically. We do not claim that CPINN is the only possible way of improving the performance of PINNs though we believe that it cuts a key reason that PINNs do not train well and provides a robust and general method that achieves greatly improved accuracy on many problems.
>
> ### Adaptive Methods and Ill-conditioning
> We agree that methods like Adam attempt to improve the conditioning by applying a diagonal rescaling. However, such a rescaling can only remove ill-conditioning that arises from singular vectors aligned with the coordinate axes. Many ill-conditioned systems have large and small singular values for which the respective singular vectors are not aligned with the coordinate axes. For instance, finite-difference discretizations of the Laplace operator are ill-conditioned, and yet their condition number can not be meaningfully reduced by a diagonal rescaling. Absent special structure, it seems unlikely that a generic ill-conditioned matrix can be well-conditioned by diagonal scaling. The fact that both PINN and CPINN require a large number of iterations also indicates that adaptive methods are not sufficient to overcome the ill-conditioning in practice.

---

### Official Review · Reviewer_hCa9 · 2022-10-21

**Confidence:** 3
**Correctness:** 4
**Technical Novelty And Significance:** 2
**Empirical Novelty And Significance:** 3
**Recommendation:** 6

**Clarity, Quality, Novelty And Reproducibility:**

The work is well-written and easy to follow. Bolded subheadings are used to create small sections, allowing for quick navigation of the paper.

The experimental quality is quite high, testing on a variety of problems with good results.

The novelty of the work is middling but non-trivial, with the addition of a discriminator network to reduce error in the PINN.

Reproducibility is possible with hyperparameters and implementational details available in the appendix.


**Strength And Weaknesses:**

Strengths:
    • Claims are supported by reported results.
    • Concept is sound and sensible.
    • Making PINNs more accurate has good impact for the field.
    • Paper is well-written and easy to read and navigate.
    • Mathematics is well explained.
    • Citations are thorough.
    • Tradeoffs between this method and conventional PINNs are discussed.

Weaknesses:
    • Figures describing the method are not included, and they would make the method easier to understand in this case.

**Summary Of The Paper:**

This paper produces more accurate physics-informed neural networks (PINNs) by including adversarial training through a discriminator which is rewarded for predicting PINN errors.

**Summary Of The Review:**

This paper seems to be a solid step forward in the field of physics-based neural networks – a field that has high impact on a number of domains. The paper is generally well-written. The novelty of the work is middling (maybe too straightforward because of the recent popularity of GANs) but non-trivial.

---

> ### Author Response · Authors · 2022-11-15
> **Response to reviewer hCa9**
>
> We thank you for taking the time to review our work and for your positive feedback. We have included a figure describing our method in the appendix, which we will move to the main part of the camera-ready version if space permits. We did not include it in the main part for now in order to keep the figure numbering constant during the discussion.

---

### Official Review · Reviewer_1E2V · 2022-10-24

**Confidence:** 4
**Correctness:** 4
**Technical Novelty And Significance:** 4
**Empirical Novelty And Significance:** 4
**Recommendation:** 8

**Clarity, Quality, Novelty And Reproducibility:**

The paper is very well-written and clear. The idea is novel, to the best of my knowledge. I cannot comment on the reproducibility as I am not sure if enough details are provided for that end.

**Strength And Weaknesses:**

Strengths:
1-	Novelty
2-	Well-written
3-	Experimental results

Weakness:
Cannot think of any


**Summary Of The Paper:**

This paper introduces a new model called CPINN, which is an agent based approach to the machine learning solution of PDEs. CPINN is an adversarial approach where the discriminator and PINN, the existing approach to solve PDEs using neural networks, plays a game in order to improve the solution precision of PDEs. The authors motivate their work by the fact that the existing PINNs approaches fail to generate precise solution. The authors show that the proposed model, CPINN, is well-behaved compared to the original PINN model, from the optimization/training perspective. The authors support their claim using multiple PDE problems.

**Summary Of The Review:**

I enjoyed reading this paper. Physics informed machine learning is a very important and hot problem and still in its infancy. The authors efforts in improving the performance of existing models for this problem needs to be highly acknowledged and rewarded. The authors were able to motivate their approach and support that using experimental study. Overall, I believe this paper has a high quality and ready for publication. I should only add that I am not fully familiar with the state of the arts in this topic and assumed that the authors reflected it well in the paper.

One minor issue: I am curious to see similar plots to that of Figure 1 for other datasets.

---

> ### Author Response · Authors · 2022-11-15
> **Response to reviewer 1E2V**
>
> We thank you for your time and the positive review. We have added heatmaps for the remaining experiments to the appendix of the revised version.

---

### Official Review · Reviewer_aueW · 2022-10-27

**Confidence:** 5
**Correctness:** 3
**Technical Novelty And Significance:** 3
**Empirical Novelty And Significance:** 3
**Recommendation:** 6

**Clarity, Quality, Novelty And Reproducibility:**

Clarity: Not clear. The work is very direct with adversarial updating point weight in the area of Neural PDE, but the terms, such as bets, game, etc., will confuse the readers.

Quality: Good. But the stop iteration is not natural. See #3 above for the detail.

Novelty: Weak. See #1 above.

Reproducibility: Good. The authors provide the code, but the method seems hard to use in other cases for requiring careful training.



**Strength And Weaknesses:**

Strength:  By introducing another neural network as the point weight function and following with a min-max optimization, the solution of the PDEs can be more accurate.

Weakness: Introducing a new neural network besides the solution neural network means much more complex for training and becomes much tricker on different tasks. And the weakness is summarized as follows.
1. there is one similar paper in 2020 as McClenny, Levi, and Ulisses Braga-Neto. "Self-adaptive physics-informed neural networks using a soft attention mechanism." arXiv preprint arXiv:2009.04544 (2020). In that work, they use mask functions for the point weight, and there is no significant difference using a neural network in this work, and the min-max procedure is the same; thus, the novelty is not much. The story is under the structure of the games but still can not change the reality of using a neural network for the point weight adversarial update. No intrinsic difference.
2. The analysis in section 2.3 is not very firm and too simple, sometimes naive, and the analysis of the neural network is just for a linear mapping which should be at least a two-layer neural network if you really want to explain something. As far as I know, in traditional numerical PDEs, the linear combination of the basis is, of course, a very common and efficient way. And it works well even using the vanilla schemes, such as the finite element method. Further, what really makes the so-called neural PDE different is the approximation by nonlinear neural networks, and it is far not enough to use one linear combination to analyze the neural PDE. There is plenty of work for analyzing PINNs, a direct search may help your analysis.
3. The results of the experiments are good but also tricky as a new neural network besides the solution neural network is introduced. More neural networks with a min-max optimization will increase the difficulties in training, and it is already complex compared with the traditional numerical solvers. If you can prove in high dimensional problem, the method still achieves good performance, just like Bao et al. in the weak adversarial neural network (Zang, Yaohua, et al. "Weak adversarial networks for high-dimensional partial differential equations." Journal of Computational Physics 411 (2020): 109409.), it is much better. Further, in the experiments, such as nearly all of the l2 error figures, the L2 relative error curve with respect to the iteration is not stable, but the experiments are just stopped at best. See Figure 4; there is still a very large oscillation. And Figure 5 has a big shock and etc. Also, it is hard to determine which one enhanced the performance, the optimizer or the structure, see Table 1.
4. The terms are a little confusing for those who are not familiar with neural PDEs and may lead misunderstandings. The work is very direct with adversarial updating point weight in the area of Neural PDE, but the terms, such as bets, game, etc., will confuse the readers. A direct statement would make things much easier.


**Summary Of The Paper:**

This paper presents an adversarial approach that overcomes this limitation, called competitive PINNs (CPINNs). CPINNs train a discriminator that is rewarded for predicting mistakes the PINN makes. The discriminator and PINN participate in a zero-sum game with the exact PDE solution as an optimal strategy. This approach avoids squaring the large condition numbers of PDE discretizations, which is the likely reason for failures of previous attempts to decrease PINN errors even on benign problems. The numerical results show the accuracy.



**Summary Of The Review:**

Based on the weakness, I reject the paper.

---

> ### Author Response · Authors · 2022-11-15
> **Response to reviewer aueW**
>
> Thank you for taking the time to read our paper. Below are our responses to your four main critiques.
> 1. The initial submission of this manuscript already referenced work similar to that of McClenny et al. in Sections 1 and 4. There, we discussed the closely related works by Wang et al. and van der Meer et al. As already described in these sections, these previous works learn positive weights to multiply with a positive loss function. Thus, the optimal strategy of the max player is always to set all weights to infinity, independent of the actions of the minimizing player. In contrast, our method using signed bets multiplied with the residual requires the maximizer to learn the mistakes of the minimizer. Thus, even if one were to use a neural network that allowed for spatially varying weights (McClenny et al. do not do this), the resulting formulation would be different. We understand that this is might be a subtle point. In order to avoid future misunderstanding, we added the above comments to section 2.2, right after introducing CPINNs.
>
> 2. As the reviewer point out, many existing works attempt to a theoretical analysis of PINNs. However, the rigorous understanding of the PINN methodology is still limited despite their being studied since 2017 (with earlier work dating back to 1998). Given this precedence, we consider a rigorous understanding of CPINNs to be beyond the scope of this paper. Instead, we focus on a model problem that motivates our approach and, we believe, cuts to the key problem associated with training PINNs. Although limited in this particular way, our theoretical contribution is in line with most other physics-informed learning papers in top-tier conferences that introduce novel methodologies and study them empirically, including those of McClenny et al. and Zang et al. mentioned by the reviewer.
>
> 3. The plots of the error vs "forward passes" account for the cost due to every single gradient computation or Hessian vector product. We thus account fairly for the additional cost incurred by competitive gradient descent, which is necessitated by the min-max formulation. Besides this cost, the general claim that "More neural networks with a min-max optimization will increase the difficulties in training" would need to be substantiated further. This has not been our experience over the course of this project and we do not see a fundamental reason why it should be the case. In ongoing preliminary experiments on high-dimensional problems, we have so far not been able to obtain meaningful improvements with the CPINN method. Doing so remains an interesting direction of future research, but our current work focuses on attaining high accuracy on low-dimensional PDEs, which comprise a large proportion of PDE applications. We point out that the only comparison of weak adversarial networks to PINNs on a high-dimensional problem in Zang et al. in Figure 2 (b), which shows almost no performance difference. We were forced to stop the training process due to the required computational resources. Note that the number of iterations and forward passes are plotted on a log scale. As shown, the experiments take a few days to train. Adding a single additional major axis tick would require training for months or longer.
>
> 4. We hope that the additional explanation contrasting bets with weights in section 2.2 will be helpful in highlighting our reasons for introducing this terminology. As described there, their difference is crucial to understanding the relationship of our method to those based on reweighting. We, therefore, consider this terminology to be useful.

---

> > ### Comment · Reviewer_aueW · 2022-11-25
> > **Concerns**
> >
> > Thanks for your reply. Here are some concerns, and I am happy if you could have further discussion.  The main concern for me is the reason for achieving high accuracy.
> >
> > 1. "Thus, even if one were to use a neural network that allowed for spatially varying weights (McClenny et al. do not do this), the resulting formulation would be different." Can you check the performance of the scheme? I mean, this would prove that the signed bets multiplied by the residual make something different. You also said the "theoretical" analysis is not rigorous and I indeed call for more proof maybe more experiments are needed.
> > 2. Okay, if it is just a not rigorous explanation, that makes sense.
> > 3. "We were forced to stop the training process due to the required computational resources.  As shown, the experiments take a few days to train."   As the computational cost is very high, an illustration of the complex problem that the traditional numerical methods can't afford would be much more reasonable, for example, complex boundary, high dimension (maybe the complex boundary problem is easier to take). Further, I am still concerned about the stopping time when the curve is still oscillating. You mentioned that because of the computational cost, it is enough to stop. Can I say that you just need good accuracy, so you can just stop at a good one? If it is, that makes sense to me.
> > 4. The terminologies are not essential for the explanation, but if you insist on contrasting bets with weights, I accept it.
> > 5. You may miss the question at the end of the third point: "Also, it is hard to determine which one enhanced the performance, the optimizer or the structure; see Table 1." I restated the question more clearly here. I mean, you compare CPINN + XCGD (X means different schemes of CGD) with PINN + Adam and SGD; is it fair?  Further, Table 1 shows if you choose CPINN + XAdam the algorithm fails, and CPINN + XSGD didn't outperform much.  That makes me confused whether the accuracy achieved is because of the XCGD.

---

> > > ### Author Response · Authors · 2022-11-29
> > > **Thank you for engaging in the discussion!**
> > >
> > > Thank you for your response. We appreciate your effort to participate in the conversation and help us improve the paper.
> > > Please see our response below.
> > >
> > > 1. To perform the requested comparison, we implemented an adaptive weighting scheme where the product of the squares of the discriminator output and the PDE residual gives the loss. As shown [here](https://imgur.com/a/8Btkp0d), we found it to perform similarly to weak adversarial networks (WAN) [Zang et al. 2019] and, thus, significantly worse than the CPINN. From a theoretical perspective, this is not so surprising because adaptive weighting is similar to WAN, though we agree with the reviewer that the experiments provided above are helpful in ascertaining this claim. The main difference between adaptive weighting and WANs is that WAN uses a logarithmic transformation of the loss, that WAN does not use adaptive weights for the boundary loss, and that the WAN loss is normalized by the magnitude of the discriminator output. This shows through both experiments and theoretical considerations that WAN and adaptive weighting are fundamentally different from the CPINN formulation, despite their superficial similarity.
> > >
> > >
> > > 2. Thank you for your understanding and for helping us to clarify the description.
> > >
> > >
> > > 3. *(Regarding comparison to traditional methods:)* We agree that finding cases where physics-informed networks can outperform classical methods is an important research direction, and that PDEs on complicated or high-dimensional domains are promising candidates. However, in this work we focus on a pervasive flaw of existing physics-informed networks that already appears in the relatively simple problems studied by a strikingly large amount of research works: Their inability to produce high-accuracy solutions even on these seemingly benign problems. We believe that resolving this limitation is a necessary and important stepping stone towards solving more challenging PDEs using neural networks and doing so performantly.
> > > *(Regarding stopping criteria:)* You are correct that, in practice, one can stop training at any point once a good solution is reached. Of course, the $L^2$ error compared to the exact solution will generally be unavailable. However, we found that the norm of the PDE residual in the training points strongly correlates with the $L^2$ error (for both PINN and CPINN). Therefore, a practical approach is to choose the iteration that achieves the lowest PDE residual. This criterion can probably be improved by including the PDE residual at "holdout points" that were not used during training. We plot the results for "as long as we could train" instead of imposing a superficial stopping criterion to provide the maximum amount of information about the training behavior of the methods under consideration.
> > >
> > >
> > > 4. Thank you for your understanding.
> > >
> > >
> > > 5. This is an important point, and we regret not addressing it more directly the first time. We provided our reasoning for why our improvements are due to the CPINN formulation and not just due to the choice of the optimizer in Section 3.6 of the revised paper. If appropriate, we are glad to expand it to the extent that space permits (or, alternatively, in an appendix).
> > > Our reasoning is as follows. The closest thing to applying BCGD to PINN would be to introduce an inconsequential "dummy agent" to obtain a problem of the form $\min_{x} \max_{y} f(x)$ and solve it with BCGD. The resulting algorithm is identical to GD. Similarly, applying ACGD to PINN is equivalent to using Adam. Indeed, the core tenet of [Schaefer and Anandkumar, 2019] is that CGD "is a natural generalization of gradient descent to the two-player setting." Thus, comparing CPINN + BCGD to PINN + GD and CPINN + ACGD to PINN + Adam strikes us as a fair comparison. As shown in Figure 6, CPINN significantly outperforms PINN in either of these cases. This justifies our claim that the improvement is due to the CPINN formulation and not just a particular optimizer. Additional evidence for this claim is provided by our experiments with WAN (in Figure 6) and adaptive weights (in our response 1. above). These experiments use ACGD, just as CPINN. Yet, CPINN achieves significantly higher accuracy.
> > >
> > > We hope that we have now addressed all your outstanding concerns. Please let us know if you require further clarification.

---

> > > > ### Comment · Reviewer_aueW · 2022-11-29
> > > > **Most of my concerns are addressed**
> > > >
> > > > Most of my concerns are addressed, thanks. "we are glad to expand it to the extent that space permits  (or, alternatively, in an appendix)" is welcomed.

---

### Author Response · Authors · 2022-11-15
**Response to all reviewers**

We thank all reviewers for reading our manuscript. We were delighted that the reviewers commented positively on the quality of our results and their exposition. We especially thank reviewers aueW and QL5E for their valuable criticism that will help us improve the paper.

In the remainder of this post, we will respond to what we perceive as the main points of critique.
We provide more detailed responses to each reviewer in a separate post.

- Reviewer aueW is concerned with the relationship of our work to the prior work that uses an adaptive method to choose the weights of the different PINN loss terms. This line of work is discussed at the end of section 4. Choosing weights for a positive loss function is fundamentally different from making bets on the sign of the residual. In the former case, the optimal strategy of the maximizing player is to move all weights to infinity. In the latter case, the optimal strategy of the maximizing player depends on the mistakes of the minimizing one. To prevent future misunderstanding, we now make this point after the introduction of CPINNs in Section 2.2.

- Reviewers aueW and QL5E suggest a comparison to weak adversarial networks (WAN) in order to better disentangle the effects of the formulation (CPINN) from the optimizer (ACGD). We have added a comparison with (WAN), trained with both Adam and ACGD, to Figure 6. We were unable to achieve accuracy comparable to CPINN with ACGD despite trying different sets of activation functions, loss functions (dropping the logarithm as suggested in section 3.5 of Zang et al.), and training modes (minibatching vs fixed batch). Nevertheless, we observe that ACGD improves the stability of WAN training. We have also added an additional paragraph to section 4 that highlights the differences between CPINN and WAN.

- Reviewers aueW and QL5E both express concern about the additional complexity introduced by the minimaximization. We regret this situation, because we believe we did account accounted for this complexity in an appropriate way, there was just a textual misunderstanding. In figures 1, 3, 4, 5, and 6, we provide the relative error as a function of not just the iterations, but also of the number of forward passes.
    The forward passes count the computation of every single gradient or Hessian-Vector product with respect to either of the networks, including those needed by the linear system solve required when computing the CGD update.
    Thus, they account for the additional computation that the CPINN method requires.

---

### Public Comment · ~Shuhao_Cao1 · 2023-02-02
**Some questions and suggestions**

Very interesting paper.

- It would be nice to see an ablation study to isolate the quantitative contributions of different components attributing to the accuracy improvement (a) the condition number change ($\kappa\to \sqrt{\kappa}$), (b) ACGD vs old schoolers (such as proximal) for saddle point problems, (c) CPINN and PINN using the same number of layers and width (in the current paper, CPINN is either wider or deeper, which translates to easier optimization according to lottery ticket).
- How a vanilla feedforward NN representation of "solution" can be written as (9) in the form of a basis expansion in terms of ***parameters*** should be elaborated in my humble opinion.

---

> ### Author Response · Authors · 2023-02-02
> **Thank you for your interest!**
>
> Thank you for your interest in our work!
>
> Regarding your first point: We determined the architectures for both PINNs and CPINNs by trying a wide range of architectures for both methods and picking whichever is best. The reason we use different architectures for the two methods is that they were their respective best-performing ones. Figure 6 provides an ablation study when using different optimizers. We observe that the classical extra gradient method on CPINNs, just like CGD, improves over PINN with SGD. These methods all use fixed and isotropic learning rates and are therefore natural to compare against one another. When introducing adaptive learning rates (XAdam, ACGD, Adam), we see that this improves performance on all methods but extra gradient, as we observe extra adam to be unstable. Our conclusion from these results is that CPINN can improve over PINN also when using other saddle point methods, such as extragradient. However, CGD is able to maintain stable training even with adaptive learning rates (ACGD), which allows it to overall achieve the best performance.
>
> Regarding your second point: We do not claim that a generic neural network representation of a solution of a PDE can be meaningfully written in the summation form described in the paper. We only point out that the classical petrov-galerkin methods can be interpreted as a very peculiar "neural network architecture", thus allowing us to relate some intuition from classical numerical analysis to (C)PINNs.

---

### Decision · Program_Chairs · 2023-01-20

**Decision:**

Accept: poster

**Justification For Why Not Higher Score:**

There are still concerns on the technical novelty - the augment to PINN is quite straightforward, although the execution of the idea looks good.

**Justification For Why Not Lower Score:**

After the discussion, the reviewers believe the ICLR audience should be interested in seeing this paper.

**Metareview: Summary, Strengths And Weaknesses:**

This paper introduces competitive physics-informed networks where two neural networks solve a partial differential equation by playing a zero-sum game. The formulation is built using a weak form. The authors claim that this weak form have smaller condition number.

The paper originally received diverse reviews, and we had a dedicated discussion among all the reviewers. The author response also helped a lot. Eventually, as a consensus, we believe this paper has more strengths than weaknesses, and it would be good to accept it to the conference.


**Note From Pc:**

if the above contains the word "oral" or "spotlight" please see: "oral" presentation means -> notable-top-5% and "spotlight" means -> notable-top-25%. As stated in our emails, we are disassociating presentation type from AC recommendations

**Summary Of Ac-Reviewer Meeting:**

QL5E:

I think this paper is different from https://arxiv.org/pdf/2009.04544.pdf in the following sense:
- the cpinn objective to solve Ax= b is min max weight *(Ax-b), the self-active pink is min max ||weight*(Ax-b)||^2
- the later formulation still suffers from the double A problem as stated in the paper
- In adaptive pinn, all the weight is positive, but in cpinn the weight can have negative,
- the introduction of two approach are different, in this paper, they intro the weight to approximate ||Ax-b||^2 using only a single A

In my opinion, we can’ t  state that both method introduce an adversarial weight thus they are similar. The mathematics behind the two papers differs.

At the same time, this paper have a nice discussion on the intuition behind the acceleration of optimization, although some of them contradicted with my own previous paper. I’m happy to see debate in this area. The claims made in author’s rebuttal also convince me the algorithm will work in some cases.

hCa9:

I read the authors' responses and others' reviews. I hold my position that this paper is 'marginally above the acceptance threshold'. The paper proposes a rather straightforward augment to PINN, which I am not particularly excited about. The execution of the ideas looks OK to me. Considering that this probably reflects the hard work of a Ph.D. student, I lean towards accepting the paper as an encouragement.

aueW:

I updated the comments two days ago and wanna see the replies from the authors. The paper achieves good accuracy, and I want to check if good results are obtained by their special design. The performance should be achieved by the difference that Reviewer QL5E mentioned, not by other reasons. To check it, some experiments of using neural networks in the adaptive PINN paper may be good proof. More importantly, there is an issue that the authors compare the results between PINN with different optimizers, and I want to clarify if it is fair. Hence I proposed several concerns in the updated review.

There is nothing wrong with Reviewer QL5E, and we all admit the difference. If it is the difference making it different, I would appreciate the novelty. There is no problem if the paper ends up being accepted as encouragement, but I want to see how they would reply to my concerns.

aueW:

The authors provided a detailed response to my latest comments and I think most of my previous concerns have been addressed.